# *Cymbopogon citratus* Essential Oil: Extraction, GC–MS, Phytochemical Analysis, Antioxidant Activity, and In Silico Molecular Docking for Protein Targets Related to CNS

**Ana G. Cortes-Torres** [1,2], **Guiee N. López-Castillo** [1], **Josefina L. Marín-Torres** [3], **Roberto Portillo-Reyes** [4], **Felix Luna** [5], **Beatriz E. Baca** [2,*], **Jesús Sandoval-Ramírez** [6,*] and **Alan Carrasco-Carballo** [1]

1   Laboratorio de Elucidación y Síntesis en Química Orgánica, ICUAP-BUAP, Puebla 72570, Mexico; alan.carrascoc@correo.buap.mx (A.C.-C.)
2   Laboratorio de Interacción Bacteria-Planta, ICCM-BUAP, Puebla 72570, Mexico
3   Herbario y Jardín Botánico Universitario, BUAP, Puebla 72570, Mexico
4   Laboratorio de Catálisis, FCQ-BUAP, Puebla 72570, Mexico
5   Laboratorio de Neuroendocrinología, FCQ-BUAP, Puebla 72570, Mexico
6   Laboratorio de Síntesis y Modificación de Productos Naturales, FCQ-BUAP, Puebla 72570, Mexico
*   Correspondence: beatriz.baca@correo.buap.mx (B.E.B.); jesus.sandoval@correo.buap.mx (J.S.-R.)

**Abstract:** This study analyzed the chemical composition of *Cymbopogon citratus* essential oil from Puebla, México, assessed its antioxidant activity, and evaluated in silico protein–compound interactions related to central nervous system (CNS) physiology. GC–MS analysis identified myrcene (8.76%), Z-geranial (27.58%), and E-geranial (38.62%) as the main components, with 45 other compounds present, which depends on the region and growing conditions. DPPH and Folin–Ciocalteu assays using the leaves extract show a promising antioxidant effect ($EC_{50}$ = 48.5 µL EO/mL), reducing reactive oxygen species. The bioinformatic tool SwissTargetPrediction (STP) shows 10 proteins as potential targets associated with CNS physiology. Moreover, protein–protein interaction diagrams suggest that muscarinic and dopamine receptors are related to each other through a third party. Molecular docking reveals that Z-geranial has higher binding energy than M1 commercial blocker and blocks M2, but not M4 muscarinic acetylcholine receptors, whereas *β*-pinene and myrcene block M1, M2, and M4 receptors. These actions may positively affect cardiovascular activity, memory, Alzheimer's disease, and schizophrenia. This study highlights the significance of understanding natural product interactions with physiological systems to uncover potential therapeutic agents and advanced knowledge on their benefits for human health.

**Keywords:** *Cymbopogon citratus*; essential oil; antioxidant; molecular docking; CNS

## 1. Introduction

*Cymbopogon citratus* is a perennial herb of the *Poaceae* family that is reproduced by cuttings. Its origin is in India, Sri Lanka, and Malaysia, and it is now cultivated in tropical and subtropical areas, including Mexico and Latin America [1]. Employed in traditional medicine worldwide, *C. citratus* has a broad range of applications, such as in antibacterial, antifungal, antiprotozoal, anticancer, anti-inflammatory, antioxidant, cardioprotective, antitussive, antiseptic, and antirheumatic activities [1–3]. Notably, it demonstrates hypoglycemic effects, making it potentially useful as an antidiabetic, and the ability to induce apoptosis in various cancer cells by increasing reactive oxygen species (ROS). Consequently, many research groups focused on elucidating the interrelation of compounds responsible for these bioactivities, whether from the complete plant or its leaves essential oil (EO). However, the action mechanisms underlying these biological effects remain largely unknown, primarily because they have been studied in isolation and without molecular objectives.

*C. citratus*, or lemongrass, exhibits a wide range of biologic activities due to its diverse bioactive compounds. The plant's antimicrobial properties encompass both antibacterial

and antifungal actions, with efficacy against Gram-negative and Gram-positive bacterial strains [4] and fungi such as *Aspergillus* spp., Mucor indicus [5], Botrytis cinerea [6], and *Aspergillus flavus* and *Aspergillus parasiticus* [7]. Lemongrass demonstrates anti-aspergillosis potential by inhibiting enzymes essential for fungal cell wall synthesis [4–8]. Lemongrass also shows promise in the realm of human health, with hepatoprotective effects against hepatocellular injury in diabetic rats [9], cholesterol-lowering potential by preventing gut absorption [10], and the ability to overcome doxorubicin resistance in cancer cells [11]. In the area of oral health, lemongrass combined with chlorhexidine can effectively reduce bacterial counts in microcosm biofilms, paving the way for new mouthwash formulations [12].

Anxiolytic, anticonvulsant, and maybe preventative qualities against Alzheimer's disease are among the plant's neuroprotective benefits that are linked to volatile components including citral, geraniol, and linalool [13,14]. As evidenced by its topical use in lowering skin erythema and edema [15] and its inclusion into chitosan bioactive films for skincare applications [16], lemongrass also has antioxidant [9,17,18] and anti-inflammatory effects. The biological activities of *C. citratus* have been attributed to the secondary metabolites present in both the plant matrix and essential oils. The species reportedly contains tannins, saponins, flavonoids, alkaloids, and, predominantly, terpenes in the leaves essential oil. Citronellal, myrcene, and *β*-pinene are of particular interest for their potential relationship with biological functions [19,20]. Recently, bioinformatic tools have contributed to establishing such a correlation.

An activity closely associated with *C. citratus* is the sedative effect due to its action on the CNS; however, this activity presents discrepancies in various studies [21]. Nevertheless, its EO does have a neuroprotective effect by promoting glutamate release and inhibiting the GABA receptor, providing a guideline for the effect among species [22]. These discrepancies could be attributed to the variants of *C. citratus* and the composition of the EO, influenced by factors such as sowing and harvesting seasons, light–dark time cycles, nutrient availability, and exposure to different herbivores, among others. Therefore, this manuscript studied the essential oil of *C. citratus* (Figure 1) grown under controlled organic conditions at the University Botanical Garden of the BUAP, using a hydrodistillation technique for the qualitative determination of its secondary metabolites, as well as quantitative analysis by GC–MS for its extraction. The antioxidant activity was assessed using the 2,2-diphenyl-1-picrylhydrazyl (DPPH) assay. Some of the obtained chemical structures were related to the reported biological activity through in silico studies of structural similarity and molecular coupling with proteins associated with CNS physiology.

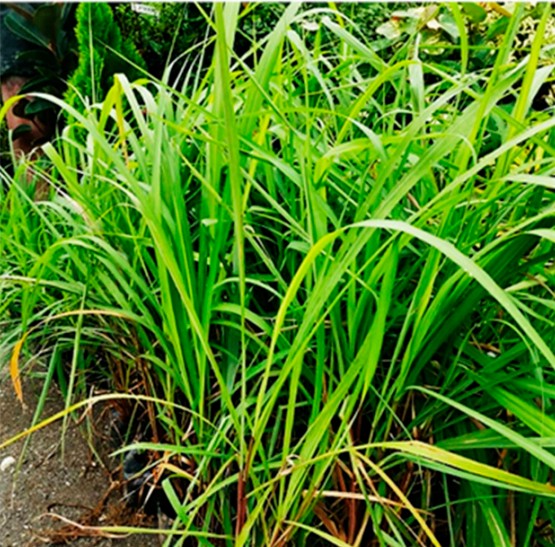

**Figure 1.** *C. citratus* of University Botanical Garden of the BUAP.

## 2. Materials and Methods

### 2.1. Vegetal Material

Fresh *C. citratus* plant material was collected between 8:00 and 10:00 am during the maturity season. It was grown under organic controlled conditions at the BUAP University Botanical Garden.

### 2.2. Extraction

The fresh plant material (100 g) was cut into 1 cm² pieces and placed in the round-bottomed flask of a modified Clevenger-type steam system, in contact with 300 mL of distilled water, and boiled in a cyclic system for 4 h. Subsequently, the volume of oil in the bubble was determined. Water was removed with $Na_2SO_4$ and then stored at 4 °C. The modification in the Clevenger equipment involved the addition of a respirator to eliminate water vapor, preventing overpressure (Figure 2).

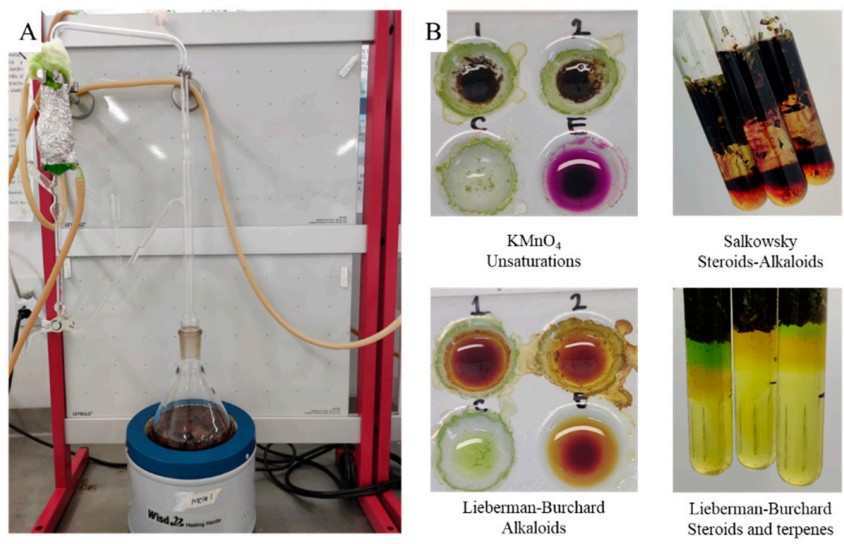

**Figure 2.** (**A**) The modified Clevenger extraction system. (**B**) Representative phytochemical tests from fresh plant material.

### 2.3. Phytochemical Test

Phytochemical tests for flavonoids, saponins, lactones, phenolic hydroxyls, unsaturation, coumarins, alkaloids, sesquiterpene lactones, steroids, terpenes, carbonyl groups, sugars, and anthocyanins were performed on both plant material and *C. citratus* EO using previously reported methodologies [23–25].

### 2.4. GC–MS Analyses

The separation, quantification, and identification of the chemical components present in the EO of *C. citratus* were carried out using Agilent Model 7890A gas chromatograph coupled with Agilent Series 5975 mass selective detector. A 20% solution of the EO (0.64 mL) in chloroform was prepared, and 1.0 μL aliquots were injected into the chromatographic column using an Agilent G4513A automatic liquid sampler. The analysis was performed in split-less mode with the injector at a temperature of 250 °C, using a fused silica HP-5MS capillary column (30 × 0.25 mm ID × 0.25 μm and He (Praxair Grade 5) as carrier gas at a constant flow of 1 mL/min. The column oven was operated using a temperature program with three heating ramps (45 °C to 150 °C @ 4 °C/min; 150 °C for 2 min; 150 °C to 250 °C @ 5 °C/min; 250 °C for 5 min; 250 °C to 275 °C @ 10 °C/min). The detector was operated in electron impact mode with an ionization energy of 70 eV and a current of 100 mA. Mass spectra were recorded in the range of 35 to 500 Da, with scan intervals of 0.32 s; the total analysis time was 71 min. The identification and calculation of relative percentage amounts of each component in the analyzed mixture were determined using

Agilent G1701EA ChemStation software library and integrators, respectively. The peaks were analyzed using NIST MS Library software.

### 2.5. Antioxidant Activity

The free radical scavenging DPPH assay was used according to the reported methodology [26,27], and absorptions were measured at 517 nm reading. The potential antioxidant activity of the essential oil was measured in triplicate samples for those obtained in independent collections. Eight solutions at different concentrations of EO in ethanol (150, 100, 75, 50, 35, 25, 10, and 1 µL/mL) were prepared. A total of 2.7 mL of EO solution was used to perform the antioxidant tests and 0.7 mL of DPPH solution. Gallic acid was used as standard and for the calculation of the percentage of inhibition the following relation was used: %Inhibition = [Blank − Test]/Blank × 100. A second test to evaluate the antioxidant activity was carried out through the determination of total phenols by the ABTS and Folin–Ciocalteu technique, according to the methodology previously reported [28–31]. The same doses mentioned above were used for the test, and the results were reported as equivalents of gallic acid.

### 2.6. In Silico Studies

The similarity study was carried out by SwissTargetPrediction, and a cumulative frequency diagram was constructed [32]. Human protein structures were obtained from the Protein Data Bank for acetylcholinesterase (AChE, 4M0E) [33], cannabinoid receptor 1 (CB1, 5TGZ) [34], cannabinoid receptor 2 (CBR2, 5ZTY) [35], D2 dopamine receptor (DR2, 6LUQ) [36], monoamine oxidase B (MAOB, 6YT2) [37], M1 muscarinic acetylcholine receptor (M1, 6ZFZ) [38], M2 muscarinic acetylcholine receptor (M2, 5ZK8) [39], and M4 muscarinic acetylcholine receptor (M4, 5DSG) [40]. All proteins were prepared in Schrödinger [41] according to the previously reported methodology, with crystallization molecules removed and water > 5.0 A, adjusting to pH 7.4 and minimizing with a RMSD < 0.3 A [42]. Reference ligands and inhibitors/blockers were obtained from the references cited in Table 3 (Pag 9) and prepared with the Schrödinger LigPrep [41] according to the previously reported methodology at pH conditions of 7.4, retaining the stereochemistry of the reported molecules and with a limit of tautomer and protonation states of 32 per molecule [42]. Finally, a non-rigid coupling system was used in Schrödinger glide [43], the softening of the nonpolar parts of the receptors was carried out by scaling the van der Waals radii by a 0.08 factor. Atoms were considered nonpolar if they were determined that their absolute partial atomic charge was <0.25, according to the methodology reported in [42], with amino acids Cys, Ser, and Tyr allowed to move freely. The slip coupling scores were performed in three high-throughput virtual sensing (HTVS) modes, standard precision (SP), and additional precision (XP). First, docking with reference molecules of the respective protein targets was performed to validate the docking protocol with OPLS4 as force field, each protein was validated by redocking with the co-crystal, obtaining RMDS < 1.5 A in all cases.

## 3. Results and Discussion

The phytochemical analysis and bioactivity of an essential oil are complex, mainly because its composition depends on several factors, such as altitude, climate, and pests, among others. Available data from different places of Mexico and other countries reveal that the composition of essential oils is not homogeneous [19,44–46]. Under controlled and completely organic cultivation conditions, carried out in the University Botanical Garden, the plant material provided a constant and accurately identified number of compounds. With this starting point, four phases of study were carried out; the first phase corresponded to the extraction and qualitative study of the essential oil's secondary metabolites. The second phase included the determination of its composition by GC–MS. The third phase involved the determination of antioxidant activity using the DPPH and Folin–Ciocalteu techniques. The fourth phase consisted of an in silico study to determine the interaction of

the found metabolites with proteins in the CNS as specific targets, to propose a hypothesis of the correlation between the essential oil's compounds and their biological function.

### 3.1. Extraction and Phytochemical Tests

The extraction process was carried out by hydrodistillation in a modified Clevenger apparatus (Figure 2A) for 4 h. In this process, the fresh plant material was always in contact with water, and the volatile compounds were extracted by steam dragging and condensed at a low temperature in a cyclic process, obtaining $0.64 \pm 0.07$ mL of oil for every 100 g of plant material.

To determine the type of secondary metabolites present in the plant material and in the oil, phytochemical tests were performed, highlighting that higher responses were obtained from the essential oil. Results of phytochemical analyses are shown in Table 1. One characteristic is that alkaloids are detected in the plant material but absent in the EO, indicating that the pathway of affectation against the CNS is not due to a psychotropic alkaloid. With the rest of the tests, congruence between plant material and EO is observed. For unsaturated and carbonyl groups, their contents increase in the essential oil, indicating that the essential oil is enriched with mono- and sesquiterpenes. On the other hand, in the case of steroids, sugars, and anthocyanins, a qualitative low presence in the EO is observed, due to the low solubility of those compounds in the water medium used for the extraction technique. Flavonoids, saponins, lactones, coumarins, and sesquiterpene lactones are not detected in either the vegetal material or the EO.

**Table 1.** Compounds identified in *C. citratus* plant material and essential oil.

| Secondary Metabolite Group | Fresh Vegetal Material | Essential Oil |
|---|:---:|:---:|
| Phenolic hydroxyls | - | - |
| Unsaturation | ++ | +++ |
| Alkaloids | + | - |
| Steroids and terpenes | +++ | + |
| Carbonyl groups | ++ | +++ |
| Sugars | +++ | + |
| Anthocyanins | ++ | - |

- no detected, + low concentration, ++ appreciable concentration, +++ high concentration.

The low amount of positive phytochemical component groups detected in fresh plant material allows for rapid analysis of EO for nonpolar and low polar compounds by GC–MS coupled to a compound library.

### 3.2. GC–MS

In the GC chromatogram, a low number of peaks can be observed compared to other essential oils and extracts of natural products, and among them, three highlighted peaks corresponding to myrcene, Z-geranial, and E-geranial at 7.7, 16.5, and 17.6 min of retention time, respectively, account for 10.7%, 32.1%, and 32.9% of the total EO content, respectively. The remaining 24.3% corresponds to 45 compounds, highlighting structures such as pulegone (0.98%, 13.28 min), (S)-linalool (1.24%, 10.69 min), tetrahydrocarvone (1.73%, 13.56 min), and geranylgeraniol (1.56%, 37.95 min). The respective structures are shown in Figure 3. These data show a high correlation with the observations in phytochemical tests. The content of secondary metabolites is quite diverse when compared to the literature. Pino et al. reported high volatile content with 33.2% Z-geranial and 39.8% E-geranial and 9.6% myrcene as major products [19]. In contrast, Ekpentong et al. reported a mixture of geranial isomers of 70–85%, as well as isopulegol, myrcene, and *L*-linalool [44]. On the other hand, Gbenou et al. reported the same components, but with myrcene at 8.63 min with 27.83% and Z-geranial at 14.25 min with 27.04%, and *E*-geranial at 15.18 min

with 19.93% [45], demonstrating that the composition and distribution of the metabolites depend entirely on the region and culture conditions. However, Pandelo et al. did report the low presence of myrcene and *Z*-geranial (31.5%) and *E*-geranial (26.1%) [46]. These reports demonstrate great variability in the content and distribution of EO components. Although geranial isomers are constant, the rest of the metabolites and their content depends on the culture conditions, and, mainly due to the biotic and abiotic stress conditions they are subjected to, the content is directly associated with various biological activities, mainly antioxidants, related to the central nervous system, and antimicrobial properties [26,44–48].

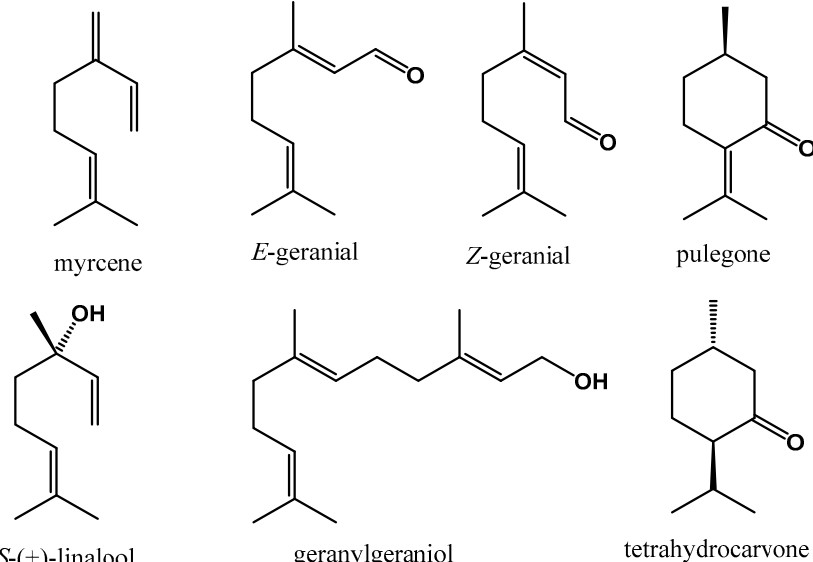

**Figure 3.** Representative phytoconstituents obtained of *C. citratus* essential oil.

### 3.3. Antioxidant Activity

Considering the presence of aldehydes, which have a latent antioxidant potential due to their reactivity, as well as the presence of some flavonoids, although in a low percentage, the antioxidant activity is viable for this essential oil. Two tests were carried out: first, the inhibitory capacity of oxidation of the DPPH free radical. Table 2 shows the percentage of inhibition obtained in relation to the concentration of the EO; one notable observation is that at 100 µL/mL in ethanol, the inhibition of the oxidative process of DPPH is better than 50%, but below 10 µL/mL, this effect is negligible. These data suggest that the most abundant metabolites are the ones that have an antioxidant effect in favor of the DPPH free radical. In the same way, in the ABTS test, a high antioxidant activity is obtained, for 150 µL EO/mL of 67.79 ± 7.55% and for 50 µL EO/mL, 45.36 ± 8.10%, demonstrating an antioxidant pathway by the terpenes contained in the oil.

**Table 2.** Antioxidant capacities of *C. citratus* essential oil.

| Concentration (µL EO/mL) | DPPH (%) |
| --- | --- |
| 150 | 76.30 ± 1.23 |
| 100 | 72.15 ± 2.62 |
| 75 | 59.58 ± 2.25 |
| 50 | 52.11 ± 1.98 |
| 35 | 18.16 ± 3.15 |
| 25 | 8.66 ± 0.87 |
| 10 | 0.0 ± 0.0 |

To determine the antioxidant level by a second test, the Folin–Ciocalteu test was carried out, and the result indicate the absence of phenols. The antioxidant activity observed at various levels demonstrates a potential effect on the CNS at the level of reducing reactive oxygen species, natural oxidants in the CNS, focusing on neuroprotective activity. Given the high lipophilicity of the compounds and the entry into the CNS, this would enable an interaction with specific proteins.

*3.4. In Silico Studies*

Given the potential physiological actions of the essential oil, both what is reported in the scientific literature and what is obtained with the antioxidant activity is of interest to study the possible interactions of the metabolites contained in this essential oil. When studying these metabolites using the SwissTargetPrediction tool, we obtained the frequency diagram shown in Figure 4, This diagram shows in percentages the frequency with which a target presents a non-zero probability of interaction of a set of molecules, in this case, the metabolites found in the essential oil. Two groups of proteins stand out; the first group is part of the hormonal system, and the second group, those of higher frequency, are associated with the neuron physiology.

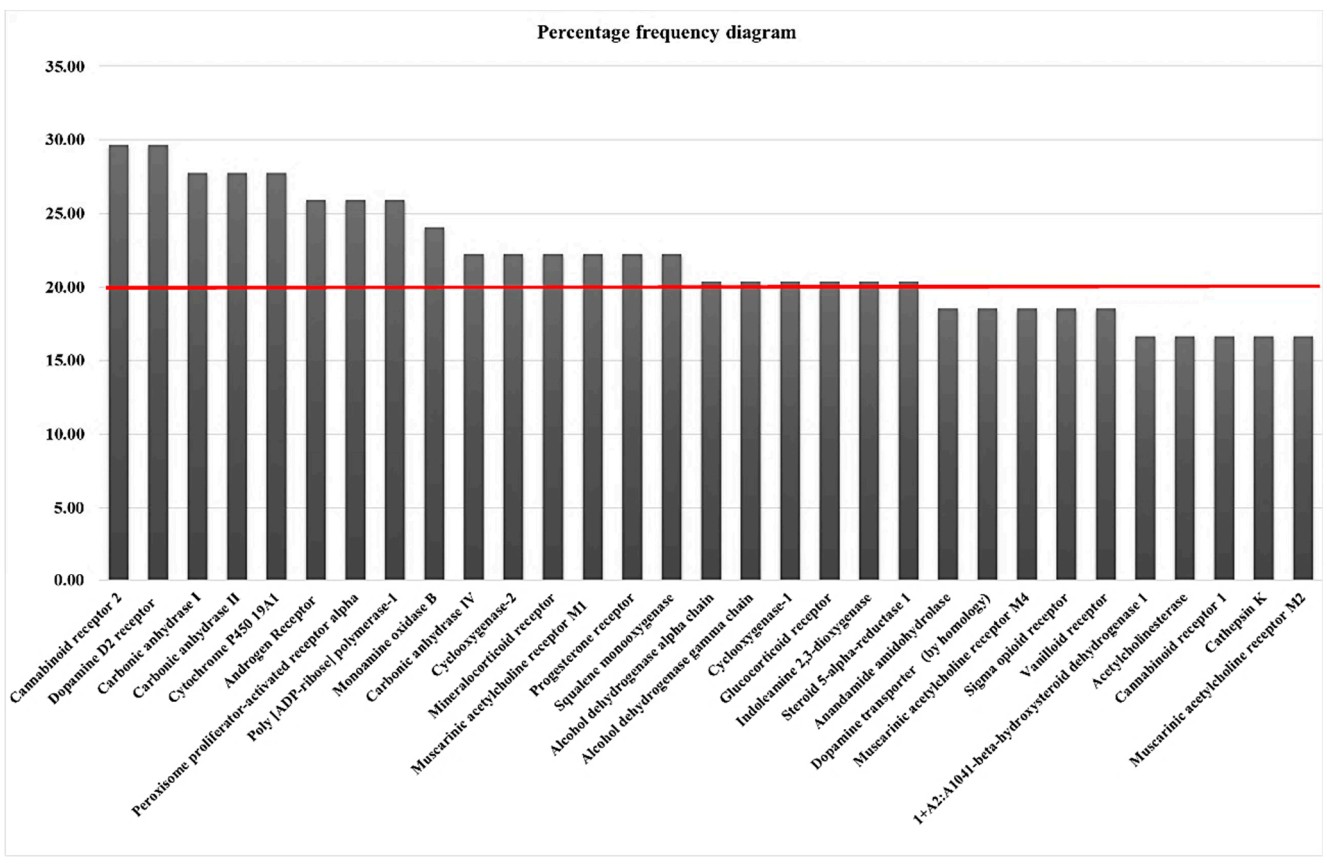

**Figure 4.** Frequency percentage diagram obtained by SwissTargetPrediction for the metabolites identified in *C. citratus* essential oil.

With the 10 proteins obtained associated with CNS physiology, a proteomic interaction diagram was constructed (Figure 5A). There is a direct interaction between these proteins at different levels. In the center is AChE [49,50], which interacts directly with the three muscarinic receptors, as it regulates the acetylcholine concentration. However, it also correlates with other proteins such as monoamine oxidase B (MAOB), dopamine receptor D2 (DRD2), cannabinoid receptor 1 (CNR1), and solute carrier family 6 member 3 (SLC6A3). Interactions between MAOB, DRD2, and SLC6A3 are possible since MAOB regulates dopamine levels and SLC6A3 regulates its transport [51]. However, the cannabinoid

receptor has no common ligands, suggesting indirect mediation with acetylcholine levels in the synaptic cleft. In the case of sigma nonopioid intracellular receptor 1 (SIGMAR1) [52,53], the interaction is correlated between cholinergic receptor muscarinic 1 (CHRM1) and DRD2, as a mediator of these two proteins. Given the obtained relationships, an interaction by third interactors is proposed, obtaining the interaction diagram with a limit of five interactors (Figure 5B), giving rise to interactions with four new proteins: guanine nucleotide-binding protein G(i) subunit alpha-1 (GNAI1), arrestin beta 2 (ARRB2), heat shock 70 kDa protein (Hsp70), binding protein 1 (HSPA), and synuclein alpha (SNCA), which explains the indirect interactions between muscarinic receptors and the dopaminergic pathway.

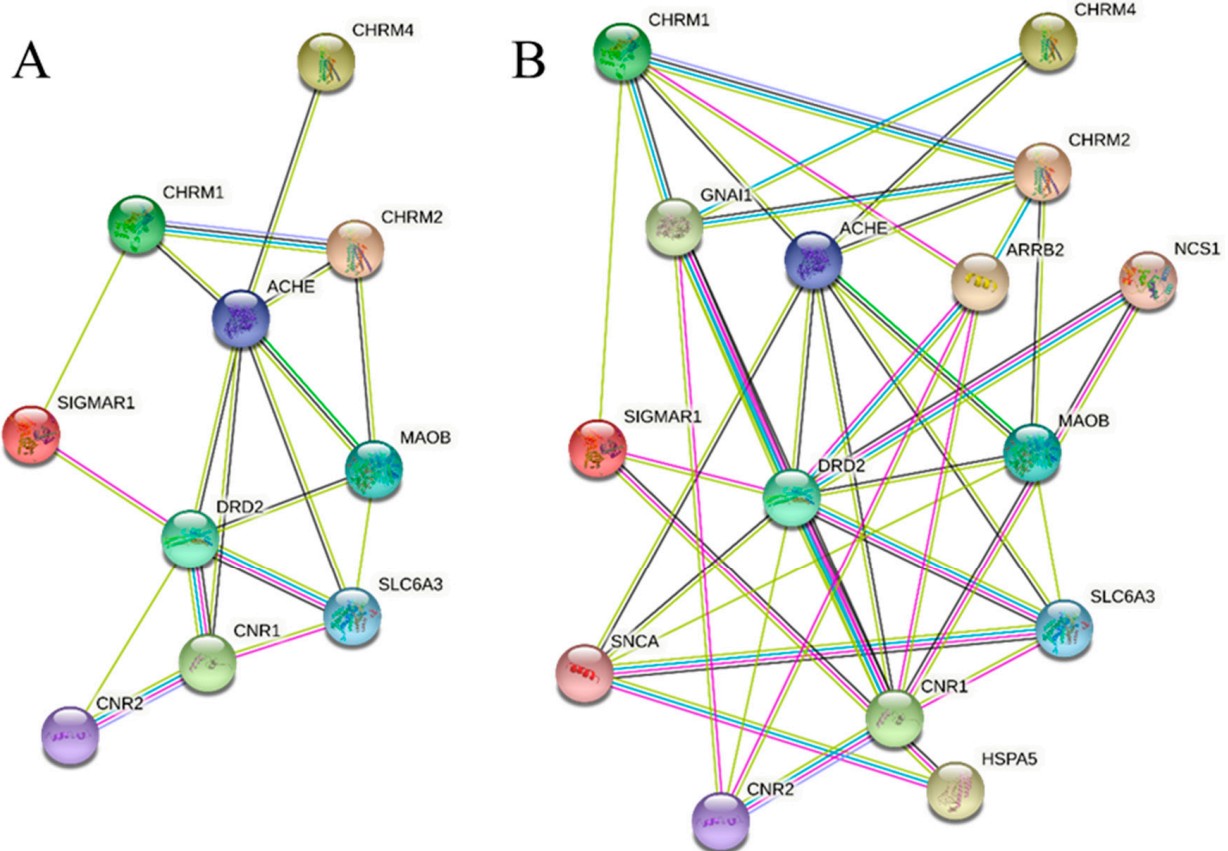

**Figure 5.** Interaction diagram between proteins by string toll: (**A**) found by STP; (**B**) found by STP and limit to five interactors.

Given the high interrelation of proteins, it was important to analyze the biological function of each of these proteins and the diseases related to them to find the possible applications of this oil and explain the effects reported in the scientific literature. Table 3 shows a summary of this information, highlighting the association with two neurotransmitters, acetylcholine [49,50] and dopamine [54–56], which play crucial roles in human physiology such as the control of voluntary movement, motivation, and cognitive functions, among others. These functions support the effect on the CNS, as altering these two systems can generate relaxation and neuroprotective effects. The pathologies related to this group of phytoconstituents are Alzheimer's and Parkinson's disease, given the regulation of acetylcholine levels and AChE activity, whose inhibition increases the levels of this neurotransmitter. In addition, an effect on cannabinoid receptors is found, which explains the effect on behavior reported for this essential oil.

**Table 3.** Target proteins, secondary metabolites in *C. citratus* and related biological function.

| Target | Frequency (%) | Substrates/Inhibitor | Disease-Related | Biological Functions | Ref. |
|---|---|---|---|---|---|
| Cannabinoid receptor 2 (CNR2) | 29.63 | JWH133, arachidonoyl ethanolamine (AEA) and 2-arachidonoylglycerol (2-AG) | Multiple sclerosis, Alzheimer's disease (AD) | Neuroinflammation, pain, and anxiety | [57,58] |
| Dopamine D2 receptor (DRD2) | 29.63 | Haloperidol, fluphenazine, chlorpromazine, risperidone, olanzapine, paliperidone, dopamine | Schizophrenia (SCZ) | Cognition, mood, and motor movements | [54–56] |
| Monoamine oxidase B (MAOB) | 24.07 | Selegiline, rasagiline | Parkinson's disease | Dopamine metabolism | [59] |
| Muscarinic acetylcholine receptor M1 (CHRM1) | 22.22 | Xanomeline | Alzheimer's disease | Learning and memory | [60] |
| Dopamine transporter (SLC6A3) | 18.52 | Dopamine, pramipexole, ropinirole, rotigotine, apomorphine | Parkinson's disease, bipolar disorder, attention deficit hyperactivity disorder, and dopamine transporter deficiency syndrome. | Translocate dopamine (DA) | [51] |
| Muscarinic acetylcholine receptor M4 (CHRM4) | 18.52 | Pirenzepine, acetylcholine | Alzheimer's disease, schizophrenia, and drug addiction | Regulation of the transmission | [40,61] |
| Sigma opioid receptor (SIGMAR1) | 18.52 | 1,3 di-*O*-tolyl guanidine (DTG), haloperidol | Addiction, depression, pain, neurodegenerative conditions, cancer, and amyotrophic lateral sclerosis | Endoplasmic reticulum stress, autophagy, lipid transport, ion channel regulation, cognition, and memory | [52,53] |
| Acetylcholinesterase (ACHE) | 16.67 | Donepezil, galantamine, | Alzheimer's disease AD, Huntington's disease, multiple sclerosis, Parkinson's disease | Development of neuromuscular junctions, thalamocortical connections, axon growth, and apoptosis | [49,50] |
| Cannabinoid receptor 1 (CNR1) | 16.67 | CP55940, JWH-015, WIN55212-2, Arachidonoyl ethanolamine (AEA), 2-Arachidonoylglycerol (2-AG) | Huntington's disease (HD), multiple sclerosis (MS), Alzheimer's disease (AD) | Learning, memory, pain, analgesia, anxiety, epilepsy, appetite | [57,62] |
| Muscarinic acetylcholine receptor M2 (CHRM2) | 16.67 | BIBN-99, 3-quinuclidinyl-benzilate, N-methyl scopolamine | Alzheimer's disease, schizophrenia, Parkinson's disease, and chronic obstructive pulmonary disease | Cardiovascular function through G-protein-coupled activation | [63,64] |

Although establishing a connection between proteins and biological function is not sufficient to predict a physiological effect, determining it involves high cost. In silico studies enable the establishment of a correlation at the energy level, allowing comparisons with endogenous ligands, substrates, or inhibitors to propose potential therapeutic effects. Table 4 presents the molecular coupling energies with each of the proteins that have a reported crystal (PDB database), as well as the energy of each of the selected controls.

**Table 4.** Coupling energies (molecular docking) among components of *C. citratus* essential oil and protein targets. Red relates to an endogenous reference. Green relates to an inhibitor reference.

| RT (min) | Name | Abundance | AChE | CBR2 | DR2 | CBR1 | M1 | M2 | M4 | MAOB |
|---|---|---|---|---|---|---|---|---|---|---|
| 7.75 | β-Myrcene | 8.76 | −4.059 | −6.017 | −5.717 | −5.840 | −6.297 | −7.025 | −6.741 | −2.987 |
| 7.87 | 4-Carene | 0.40 | −4.456 | −6.637 | −5.897 | −6.341 | −6.787 | −7.040 | −7.276 | −3.390 |
| 8.56 | β-Pinene | 1.94 | −4.709 | −6.193 | −5.459 | −5.827 | −6.163 | −6.762 | −6.931 | −3.173 |
| 9.07 | α-Ocimene | 0.40 | −1.969 | −3.778 | −3.422 | −3.315 | −3.196 | −4.042 | −3.971 | −0.510 |
| 9.18 | β-Ocimene | 0.40 | −2.376 | −4.469 | −4.005 | −4.170 | −3.947 | −4.691 | −3.841 | −0.932 |
| 9.41 | Seudenone | 0.07 | −4.584 | −5.897 | −5.906 | −4.940 | −6.193 | −6.324 | −6.312 | −3.512 |
| 10.42 | 1,2,3,3a,4,6a-hexahydropentalene | 0.20 | −4.013 | −5.900 | −5.607 | −5.511 | −6.106 | −6.403 | −6.428 | −2.949 |
| 16.89 | d-Linalool | 1.24 | −2.649 | −4.713 | −3.538 | −4.894 | −4.478 | −4.332 | −4.822 | −1.569 |
| 11.52 | Alloocimene | 0.38 | −3.153 | −5.191 | −4.324 | −5.478 | −5.089 | −5.323 | −5.404 | −1.780 |
| 12.05 | 3,3,5-Trimethyl-1,4-hexadiene | 0.31 | −2.900 | −4.449 | −4.072 | −4.109 | −4.443 | −4.905 | −4.628 | −1.855 |
| 12.27 | 3,5-Dimethyl-1,6-heptadiene | 0.80 | −1.027 | −2.954 | −2.231 | −1.742 | −2.830 | −2.724 | −3.042 | 0.212 |
| 12.69 | Ethenylcyclohexane | 0.81 | −4.425 | −5.844 | −5.079 | −5.891 | −6.138 | −6.510 | −6.508 | −3.306 |
| 12.96 | 5-Dodecyne | 0.98 | 1.056 | −0.991 | −0.398 | 0.796 | 0.173 | −1.252 | −1.262 | 3.277 |
| 13.29 | Pulegone | 1.00 | −4.594 | −6.959 | −6.407 | −6.483 | −6.581 | −7.366 | −7.774 | −3.612 |
| 13.56 | Tetrahydrocarvone | 1.73 | −4.813 | −6.692 | −5.646 | −5.456 | −6.533 | −6.429 | −7.075 | −3.061 |
| 15.76 | Z-Geranial | 27.58 | −2.263 | −4.050 | −4.437 | −4.384 | −4.935 | −4.555 | −4.756 | −1.062 |
| 17.7 | E-Geranial | 38.62 | −2.787 | −4.535 | −4.359 | −5.533 | −4.187 | −4.679 | −4.793 | −1.283 |
| 18.18 | 3-Ethoxybenzenamine | 1.23 | −4.391 | −5.220 | −5.261 | −5.832 | −5.407 | −5.257 | −5.438 | −3.109 |
| 18.37 | Grandlure IV | 0.22 | −4.029 | −6.262 | −5.123 | −5.166 | −5.782 | −7.163 | −6.447 | −2.975 |
| 18.73 | Carvotanacetone | 0.48 | −5.141 | −7.020 | −6.368 | −6.551 | −7.282 | −7.194 | −7.761 | −3.574 |
| 18.99 | Ethyl nerate | 0.29 | −3.835 | −5.454 | −4.512 | −6.495 | −4.719 | −5.766 | −5.264 | −1.607 |
| 19.13 | Verbenyl ethyl ether | 0.74 | −4.573 | −6.224 | −4.924 | −5.807 | −6.031 | −7.097 | −7.071 | −2.783 |
| 19.75 | 2,3-Dimethyl-3-buten-2-ol | 0.76 | −3.739 | −4.383 | −3.571 | −4.536 | −4.426 | −4.825 | −4.758 | −2.254 |
| 20.26 | (Z)-2-Butenoic acid, methyl ester, | 1.33 | −3.615 | −3.541 | −3.838 | −3.949 | −4.181 | −3.854 | −4.308 | −2.088 |
| 20.62 | Dehydrolinalool | 0.22 | −2.662 | −4.383 | −3.847 | −4.739 | −4.288 | −4.654 | −4.763 | −2.590 |
| 20.85 | 2-Butenoic acid, hexyl ester | 0.77 | 1.204 | −0.796 | −0.220 | −1.109 | 0.259 | −0.893 | −0.726 | 2.892 |
| 21.03 | Chrysanthenone | 0.30 | −5.398 | −6.865 | −4.983 | −6.817 | −6.342 | −7.531 | −7.678 | −3.517 |
| 21.12 | Safranal | 0.30 | −5.559 | −6.299 | −5.963 | −5.861 | −6.250 | −7.021 | −7.470 | −3.710 |
| 21.47 | Butyl crotonate | 0.61 | −2.258 | −3.482 | −2.791 | −3.802 | −3.056 | −3.889 | −4.022 | −0.556 |
| 21.83 | (Z)-3,7-Dimethyl-2,6-octadienal | 0.08 | −2.263 | −4.050 | −4.437 | −4.384 | −4.935 | −4.555 | −4.756 | −1.062 |
| 23.12 | 2-Tridecanone | 0.81 | 0.890 | −1.035 | −0.113 | −1.947 | −0.851 | −1.690 | −1.094 | 2.938 |
| 25.68 | Caryophyllene oxide | 0.07 | −4.565 | −6.944 | −4.517 | −6.219 | −3.925 | −8.261 | −7.851 | −2.936 |
| 26.67 | α-Guajene | 0.09 | −5.065 | −7.370 | −6.424 | −6.932 | −6.372 | −8.089 | −8.295 | −3.151 |
| 28.64 | 2-Pentadecanone | 0.05 | 0.437 | −1.584 | 0.262 | −2.187 | −0.710 | −1.950 | −1.508 | 2.572 |
| 33.72 | 2,2,3,5,6-Pentamethyl-3-heptene | 0.03 | −4.172 | −6.157 | −5.614 | −5.976 | −5.908 | −6.434 | −6.610 | −1.934 |
| 37.31 | 1-Benzyloxy-9-(phenylthio)-3,7,11,15-tetramethyl-2,6,10,14-hexadecatetraene | 0.22 | −3.920 | −10.474 | −4.460 | −9.091 | −7.890 | −9.524 | −8.581 | −1.828 |
| 37.94 | Geranylgeraniol | 1.56 | −2.125 | −4.829 | −4.066 | −4.879 | −4.051 | −4.577 | −4.736 | −0.007 |
| 38.36 | 2-cis-Geranylgeraniol | 0.27 | −2.250 | −4.800 | −3.307 | −5.289 | −4.246 | −4.117 | −4.591 | 1.017 |
| 38.6 | 2,6-Dimethyl-6-(4-methyl-3-pentenyl)-2-cyclohexene-1-carboxaldehyde | 0.48 | −4.219 | −6.127 | −5.912 | −6.670 | −6.351 | −7.792 | −7.059 | −2.108 |
| 38.76 | Geranyllinalool | 0.52 | −1.561 | −4.981 | −3.221 | −5.242 | −4.102 | −4.056 | −5.000 | 0.133 |
| 39.18 | 2,6,10,15,19,23-Hexamethyl-2,6,10,14,18,22-tetracosahexaene (all-E) | 1.39 | −4.129 | −8.479 | −5.427 | −7.232 | −6.022 | | −7.048 | −1.266 |
| 39.47 | 2,6,10,14,18-Pentamethyl-2,6,10,14,18-eicosapentaene | 0.85 | −4.031 | −8.025 | −3.890 | −7.795 | −5.945 | −4.998 | −7.618 | −1.129 |
| 40.05 | Peruviol | 0.26 | −1.133 | −4.067 | −1.594 | −3.935 | −2.811 | −3.895 | −4.044 | 1.063 |
| 40.24 | Lavandulol | 0.13 | −2.795 | −4.559 | −4.117 | −4.453 | −4.722 | −5.130 | −5.083 | −2.489 |
| 41.14 | 2-Methyl-2-(4-methyl-3-pentenyl)cyclopropanemethanol | 0.11 | −3.923 | −5.160 | −4.973 | −5.131 | −6.096 | −5.762 | −5.946 | −2.383 |
| 41.97 | 4,8-Dimethyl-3,7-nonadien-2-ol | 0.06 | −4.114 | −5.556 | −4.824 | −5.864 | −5.379 | −5.578 | −5.993 | −2.674 |
| - | ligand/substratum | * | −3.934 | −7.224 | −8.16 | −5.107 | −4.19 | −4.484 | −6.04 | - |
| - | blocker/inhibitor | + | −5.459 | −8.085 | −9.27 | −7.686 | −4.613 | −9.538 | −4.665 | −5.766 |

RT = retention time. * Acetylcholine: AChE, M1, M2 and M4; CBD: CBR1 and -CBR2; dopamine: DR2. + AChE: donepezil, M1: Xanomeline, M2: methyl scopolamine, M4: pirenzepine, CBR1 and CBR2: THC, DR2: haloperidol, MAOB: rasagiline.

Initially, inhibition or a direct effect on the dopamine receptor appears unlikely since all molecules have lower energy than dopamine. Under physiological conditions, dopamine displaces any of them, reinforcing the theory of indirect interaction through the acetylcholine pathway. Two possible blocks of effects can be suggested: the first involves direct action on the muscarinic receptors M1, M2, and M4. Most metabolites can block these three receptors with better energy than the endogenous ligand. In M1, they exhibit higher energies than the reported blocker (Ki = 4 nM). Specifically, geranial (in higher abundance) can block M1 (better than the commercial blocker) and M2 (Ki = 31 nM) (better than acetylcholine) but not M4, however, this receptor shows an effect at different levels, presenting Ki values from 13.8 to 105.0 [65,66]. Meanwhile, pinene and myrcene, potential tertiary components in content, can block all three receptors, leading to deregulation in cardiovascular physiology and memory. However, in the case of myrcene, this action could affect the corticosteroid pathway, which might be beneficial in Alzheimer's disease and schizophrenia [65], but the effect on cardiovascular functioning is negative. The downregulation of acetylcholine levels is regulated by the observed effect as inhibitors of acetylcholinesterase activity since the E-geranial can inhibit this enzyme, although its isomer does not have this potential effect. Myrcene and pinene and other metabolites contained in the essential oil, including safranal, show better binding energy than reference inhibitor donepezil.

On the other hand, for MAOB, no metabolite has higher energy than rasagiline (IC$_{50}$ = 0.042 mg/kg). The inhibition of this enzyme involves the metabolism of dopamine, which clarifies the interactions observed in the protein interaction diagrams. Figure 6 shows the interactions that phenelzine presents compared with the compounds with the highest abundance, with unions with Asn145 and Glu150. Specifically, *E*- and *Z*-geranial do not show significant interactions with these two amino acid residues but with Lys149 close to the site.

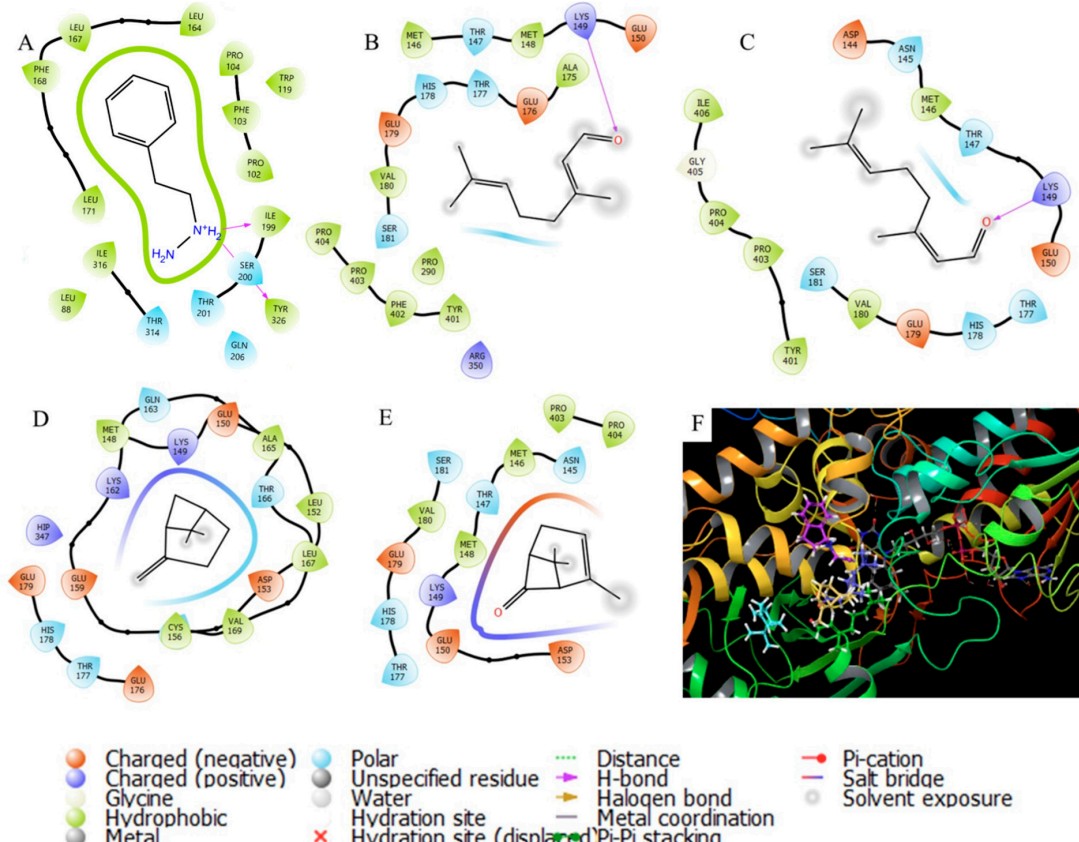

**Figure 6.** Molecular interactions among MAOB enzyme and essential oil compounds of *C. citratus*: (**A**) phenelzine; (**B**) *E*-geranial; (**C**) *Z*-geranial; (**D**) β-pinene; (**E**) pulegone; (**F**) 3D interactions.

The third associated group is the cannabinoid receptors, which are mainly associated with response in the CNS given the psychotropic and relaxant effects. Terpenes have a lipophilic structure similar to CBD and THC [57,58,62]; however, there is a greater interaction energy on CBR1 (THC, Ki = 10 nM) than on CBR2 (THC, Ki = 24 nM). Myrcene and *E*-geranial can interact on the CBR2 receptor type 1 [57], while with type 2, they do not reach the level of CBD. These data suggest that the psychotropic effects occur through the CBR1 pathway. Figure 7 shows the interactions they have with each of the metabolites, both CBD and THC, as well as the metabolites with higher content, maintaining the binding in the same place since the interactions are observed with the same residues. The type of this bond generates changes in the coupling energy between the E and Z-geranial due to the interaction distance with Met103, unlike CBD, which does not present said energy. The molecule with the highest coupling energy has the disadvantage of having an abundance of 0.22%, which makes it difficult to have a more significant effect than THC. In the case of high abundance on CBD, it suggests that the relaxing and psychotropic effect could be explained by the activation of CBR1 receptors but not by the CBR2 receptors, which explains why, in some countries, the use of this natural product is regulated.

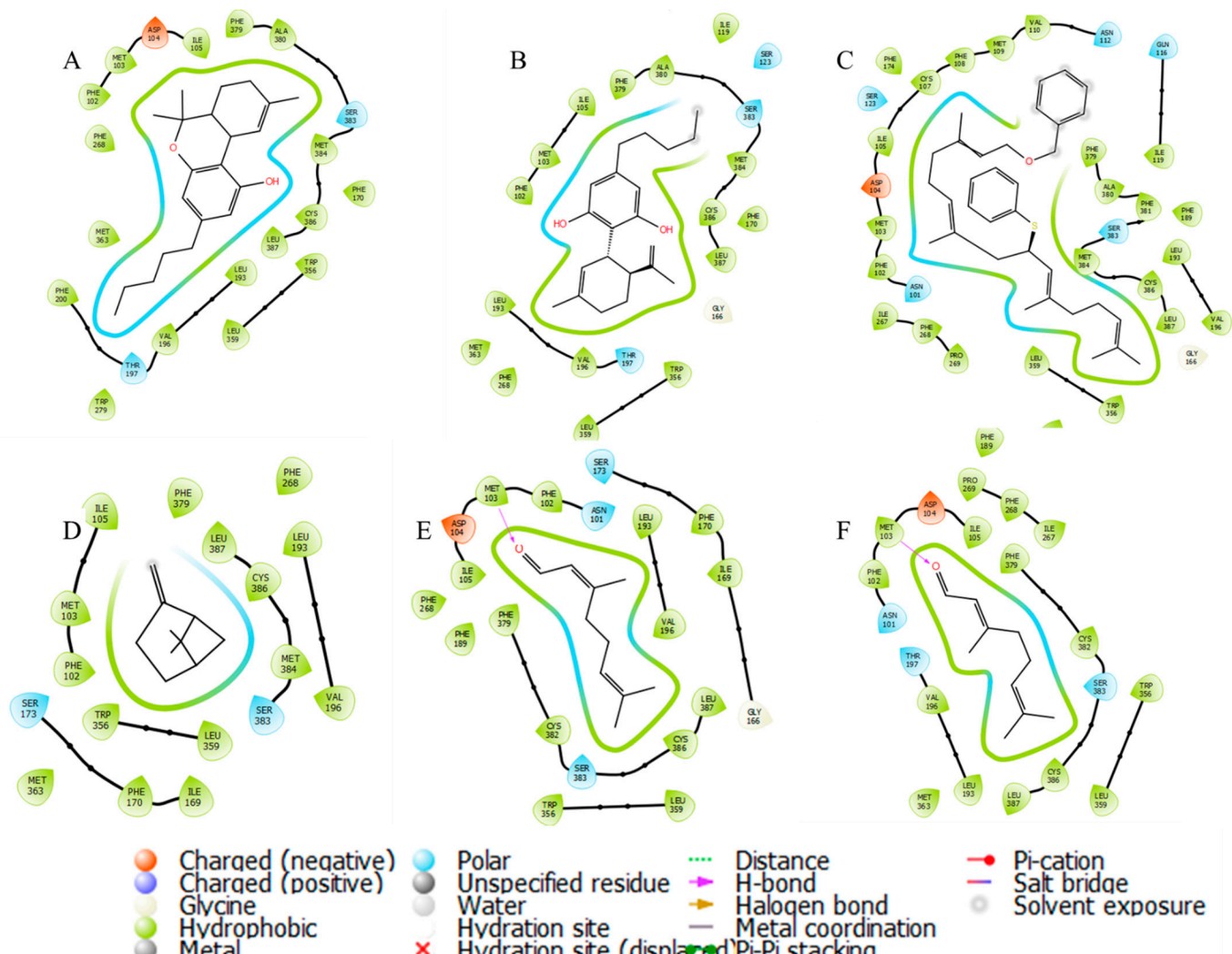

**Figure 7.** Molecular interactions among CBR1 receptor and essential oil compounds of *C. citratus*: (**A**) THC; (**B**) CBD; (**C**) TL-37-53; (**D**) β-pinene; (**E**) *Z*-geranial; (**F**) *E*-geranial.

## 4. Conclusions

This study elucidated the chemical composition, antioxidant activity, and potential physiological effects of *Cymbopogon citratus* essential oil. The presence of various phytochemical components suggests possible therapeutic applications, while in silico analysis reveals interactions with central-nervous-system-associated proteins. The essential oil may affect the central nervous system through multiple pathways, including cholinergic and dopaminergic systems and cannabinoid receptor type 1 (CBR1) activation. This could account for the reported psychotropic and relaxing effects and potential therapeutic applications in neurological conditions.

Although in silico analyses offer valuable insights, further experimental, in vivo, and clinical studies are required to confirm the interactions and efficacy of the essential oil in treating various diseases and conditions. Ultimately, this research enhances our understanding of the essential oil's chemical composition and potential therapeutic applications, emphasizing the importance of exploring complex interactions between natural products and physiological systems to discover novel therapeutic agents and advance our knowledge of these compounds' potential benefits for human health.

**Author Contributions:** Conceptualization, A.C.-C., J.S.-R. and B.E.B.; methodology, A.G.C.-T. and G.N.L.-C, validation; A.C.-C., F.L. and J.S.-R.; formal analysis, A.C.-C., F.L. and R.P.-R.; investigation, A.G.C.-T., G.N.L.-C and A.C.-C.; resources, J.L.M.-T., J.S.-R. and R.P.-R.; data curation, A.C.-C.; writing—original draft, A.G.C.-T. and A.C.-C.; writing—review and editing, J.S.-R., B.E.B. and A.C.-C.; visualization, J.L.M.-T. and G.N.L.-C.; supervision, A.C.-C. and J.S.-R.; project administration, A.C.-C. and J.S.-R.; funding acquisition, J.S.-R., A.C.-C. and B.E.B. All authors have read and agreed to the published version of the manuscript.

**Funding:** The research project was supported by CONACYT by project Pronaces-317580, and VIEP-BUAP.

**Institutional Review Board Statement:** Not applicable.

**Informed Consent Statement:** Not applicable.

**Data Availability Statement:** All data are available in the manuscript.

**Acknowledgments:** Villegas R. for English and style corrections.

**Conflicts of Interest:** The authors declare no conflict of interest.

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
