# Peer review of "Cymbopogon citratus Essential Oil: Extraction, GC–MS, Phytochemical Analysis, Antioxidant Activity, and In Silico Molecular Docking for Protein Targets Related to CNS"

_cimb, doi:10.3390/cimb45060328_

Round 1

Reviewer 1 Report

It is a well-organized article and without major errors. I am impressed by the work done and the quality of the manuscript. The article was presented in a very good and accurate way. The methodology section is at an appropriate scientific level. In addition, the work is written very carefully, and the language is good. Therefore, I recommend it for publication. However, certain Minor issues are detailed below which need to be addressed before its final acceptance.

I advise the authors to take the following points into account while revising their manuscript.

1. The authors that I found there are some typographical errors in the manuscript, so authors need to correct them in the revised manuscript.

2. Let the author focus main points and explain the research question clearly. The abstract section should be revised.

3. Include the Graphical Abstract in the revised manuscript to attain a broad readership.

4. I think that part about in silico study could be extended. This would be valuable for later publication citation.

5. The content of the volatile metabolites in the essential oil is variable according to the  cultivation conditions should be more emphasized.

6. The manuscript should be extended in scientific discussion. The authors presented their results and compared to some works, but did not present explanations for the reasons to reach these results.

7. Not all of the described results are covered in the discussion section. 

8. Revise the conclusion section with future prospectives and also conclusions section should be elaborated.

9. No all information was given about biological activities of C. citratus.

Author Response

It is a well-organized article and without major errors. I am impressed by the work done and the quality of the manuscript. The article was presented in a very good and accurate way. The methodology section is at an appropriate scientific level. In addition, the work is written very carefully, and the language is good. Therefore, I recommend it for publication. However, certain Minor issues are detailed below which need to be addressed before its final acceptance.

I advise the authors to take the following points into account while revising their manuscript.

  1. The authors that I found there are some typographical errors in the manuscript, so authors need to correct them in the revised manuscript.

The entire document has been reviewed to correct typographical errors, and it has been sent for style and language correction.

  1. Let the author focus main points and explain the research question clearly. The abstract section should be revised.

The Abstract has been restructured, the introduction modified to enhance the importance of this study, and its approach explained otherwise.

  1. Include the Graphical Abstract in the revised manuscript to attain a broad readership.

The GA was already included, lighting and emphasizing the critical stages of this study, as well as the results associated with CNS.

  1. I think that part about in silico study could be extended. This would be valuable for later publication citation.

Discussion on results associated with cannabinoid and muscarinic receptors has been extended to relate to reported effects as well as possible effects associated with AChE and MAO enzymatic inhibition.

  1. The content of the volatile metabolites in the essential oil is variable according to the cultivation conditions should be more emphasized.

The composition changes according to the growing area have been emphasized in the discussion, conclusions, and in content, including references from studies in other regions of the world.

  1. The manuscript should be extended in scientific discussion. The authors presented their results and compared to some works but did not present explanations for the reasons to reach these results.

A relationship between the obtained results has been included and emphasized in the conclusion section for the most relevant items as well as emphasizing the set of characterized metabolites.

  1. Not all of the described results are covered in the discussion section. 

The discussion has been enlarged for each of the in silico results, as well as those associated with antioxidant results.

  1. Revise the conclusion section with future prospectives and also conclusions section should be elaborated.

The conclusions have been restructured, including the perspectives of this type of study and emphasizing the relative content of metabolites according to the area and cultivation conditions, this being the main value to continue studying this plant species.

  1. No all information was given about biological activities of C. citratus.

More references have been included to increase the bibliographical heritage on previously reported biological activities towards the essential oil of C. citratus.

Reviewer 2 Report

Dear Editor,

I cannot recommend publication of the paper in its present form.

First of all, the language is plagued by several errors, some of which apparently scattered carelessly and abundantly. Worse, some construction errors affect the text comprehension.

I list some observations I collected:

- myrcene appears twice in the GC-MS composition described in the abstract, the sum of percentages results therefore odd.

- SwissTarjetPrediction should be SwissTargetPrediction

- line 23: targets of what?

- line 77: "Water was removed with Na2SO4 and then stored at 4 °C", this does not mean what the authors intend, I guess

- lines 107, 108: what an odd construction!

- line 126: "Try" for "Tyr". but which aminoacids where left non rigid is not clear to me.

- lines 131, 132: their oil?

- line 137: "GS-MS"

- lines 193, 194: ??

- lines 204-209: another hardly readable phrase, too many errors to list.

- how would the antioxidant test data support the conclusion that the "most abundant metabolites are the ones that have an antioxidant effect in favor of the DPPH free radical"?

- I would double check the results for binding energies (and mention the measurement units). For example, the binding energy between M1 and its reference xanomeline (why?) is reported by the authors as -4.613 kcal/mol, I guess, however this is a long way offset from the experimental values of Kd, which are between 8 and 3000 nM, and consequently correspond to energies between -11 and -7.5 kcal/mol. Still more suspect is the binding energy between MAOB and its reference rasagiline, which, at least, would make rasagiline an unwise reference, instead its binding constant is reported at different places in the literature, with values between 7 and 700 nM.

It is my guess that the authors have poor acquaintance with docking and its interpretation. I suggest that they review the adopted methodology or at least their confidence in such results.

The authors should also explicit what they mean with the green cells of table 4: I initially thought those were the values better then the reference ones but I had to change idea after looking at the last two columns.

It really appears that the paper has been assembled in a hurry. Please review with great care. Some amendments are suggested above.

Author Response

  1. First of all, the language is plagued by several errors, some of which apparently scattered carelessly and abundantly. Worse, some construction errors affect the text comprehension.

I list some observations I collected:

- myrcene appears twice in the GC-MS composition described in the abstract, the sum of percentages results therefore odd.

- SwissTarjetPrediction should be SwissTargetPrediction

- line 23: targets of what?

- line 77: "Water was removed with Na2SO4 and then stored at 4 °C", this does not mean what the authors intend, I guess

- lines 107, 108: what an odd construction!

- line 126: "Try" for "Tyr". but which aminoacids where left non rigid is not clear to me.

- lines 131, 132: their oil?

- line 137: "GS-MS"

- lines 193, 194: ??

- lines 204-209: another hardly readable phrase, too many errors to list.

The entire document has been reconsidered to correct typographical errors, and content. The manuscript has been sent for style and language correction to a specialized department in the university.

  1. how would the antioxidant test data support the conclusion that the "most abundant metabolites are the ones that have an antioxidant effect in favor of the DPPH free radical"?

The DPPH radical scavenging inhibition study allows quantifying the antioxidant potential of a substance or a mixture since it neutralizes the oxidizing effect of this radical. This same activity may be due to various known antioxidants, some of those found in the GC-MS study; that is the case for terpenes, which have previous reports of antioxidant activity. Their references are included, as well as we have added ABTS test to reinforce support for such antioxidant potential.

  1. I would double check the results for binding energies (and mention the measurement units). For example, the binding energy between M1 and its reference xanomeline (why?) is reported by the authors as -4.613 kcal/mol, I guess, however this is a long way offset from the experimental values of Kd, which are between 8 and 3000 nM, and consequently correspond to energies between -11 and -7.5 kcal/mol. Still more suspect is the binding energy between MAOB and its reference rasagiline, which, at least, would make rasagiline an unwise reference, instead its binding constant is reported at different places in the literature, with values between 7 and 700 nM.

The reference inhibitors/blockers have been reviewed and updated in each case. The color code used to compare the metabolites which have better binding energies than the endogenous substrate/ligand and those that are better than the reference inhibitors/blockers has been marked. Regarding a direct relationship between the docking score and the proposed Kd, this is not directly proportional since the final experimental value involves processes in the distribution membrane, solubility, among others, so the in silico study only allows the prediction of potential interaction. Phenelzine has been updated as a MAOB inhibitor, its reference is added.

  1. It is my guess that the authors have poor acquaintance with docking and its interpretation. I suggest that they review the adopted methodology or at least their confidence in such results.

The methodology used is duly validated; the discussion for the in silico section, and its interpretation has been improved. The description in the methodology has been extended to be explicit.

  1. The authors should also explicit what they mean with the green cells of table 4: I initially thought those were the values better then the reference ones but I had to change idea after looking at the last two columns.

An explanation of the color code used has been included as well as more discussion regarding the obtained results.

Round 2

Reviewer 2 Report

Dear Editor,

the paper has been improved but I do not deem it ready for publication.

First of all, another pass of careful reading from the authors would have been beneficial, would one really believe that the paper starts with "This in this work the"? I know the paper underwent revision by professionals, and I am not an English specialist, yet I can assure that it has still some problems, probably where the specialist could not well understand the intent of the authors. Below some sample observations.

-l153 predators feed on animals, not plants.

- between 8:00 to 10:00 am should be between 8:00 and 10:00 am

-l311 according to the previously reported methodology (the authors probably mean: according to the methodogy reported in [27]. it is not even clear why the authors cite themselves, did they establish such methodologies first?)

-l311 missing link to the following text

-l389 qualitative decrease: what does it mean?

-l413 However, Pandelo et al. did not report the presence of myrcene and a lower percentage of Z-geranial: certainly the authors mean that Pandelo reports a lower percentage of Z-geranial, not that he does not mention it.

-l472 result indicated no phenols present: "no phenols are present" or "the absence of phenols"

-l475 Another possibility is the high lipophilicity of the compounds and the entry into the CNS,
which would enable an interaction with specific proteins. reformulate, by the way "given the high lipophilicity" was a good part of the sentence.

- figure 4 and related text: frequency and percentage of what? the mechanics of how they used STP is not clear

-l573: Table 4 presents the molecular coupling energies with each of the proteins that had a reported crystal (PDB database) are shown,: too many verbs here!

-l723: however. : however what?

However, this is not the main problem with the paper. Outstandingly, in their answer to my objections, the authors seem to understand my point about table 4, however this empathy does not make itself into the paper. Instead they go at length into discussing how table 4 purports a long range of quantitative deductions: what would displace what and so on, with no appreciation or mention of the accuracy of the reported data or comparison with available experiments. May I at least make them notice at least three oddities?

1. most of the compounds in their oil bind M1 better than the reference

2. worse, more than half the compounds in their oil bind at least 5 receptors more than the reference ones. is this the receipt for a kind of panacea or a poison?

3. geranial, shall we really discuss it? At any rate I tested it on STP and even other similar software like PLATO and they do not give any credit to it as a plausible M1 target: actually M1 is at the seventh place and no target anyway exhibits a significant probability that geranial is a binder of interest.

There are interesting data in the paper, and even some potentially interesting compounds like 1-Benzyloxy-9-(phenylthio)-3,7,11,15-tetramethyl-2,6,10,14-hexadecatetraene. The problem is that the conclusions are not warranted by the methods, and this is mostly due to the authors' excessive reliance on their numbers. I am not able to discern the origin of such errors, but it is rather clear that even the relative error in the binding energies intra columns is at least 2 kcal/mol. Certainly it is much more among the different columns, where some data are in line thermodynamically with experimental data while others are off by at least 4 kcal/mol, as I showed in the first reviewing round. It is easy to spot, for example in the large difference of binding energies in the references for M1 and M2.

Included in general comments.

Author Response

We appreciate the reviewer's comments made, below you will find the response assigned to each of your comments, as well as the modifications made to the manuscript, with change control activated.

First of all, another pass of careful reading from the authors would have been beneficial, would one really believe that the paper starts with "This in this work the"? I know the paper underwent revision by professionals, and I am not an English specialist, yet I can assure that it has still some problems, probably where the specialist could not well understand the intent of the authors. Below some sample observations.

The English was revised again after the corrections made.

-l153 predators feed on animals, not plants.

- between 8:00 to 10:00 am should be between 8:00 and 10:00 am

-l472 result indicated no phenols present: "no phenols are present" or "the absence of phenols"

These points were directly corrected.

-l311 according to the previously reported methodology (the authors probably mean: according to the methodogy reported in [27]. it is not even clear why the authors cite themselves, did they establish such methodologies first?)

The wording was corrected and the methodology was explained in detail.

-l311 missing link to the following text

-l389 qualitative decrease: what does it mean?

-l413 However, Pandelo et al. did not report the presence of myrcene and a lower percentage of Z-geranial: certainly the authors mean that Pandelo reports a lower percentage of Z-geranial, not that he does not mention it.

-l475 Another possibility is the high lipophilicity of the compounds and the entry into the CNS,
which would enable an interaction with specific proteins. reformulate, by the way "given the high lipophilicity" was a good part of the sentence.

The wording has been corrected to present greater clarity, focused on what was reported as well as focusing on high lipophilicity as a guide to the possible effect on CNS.

- figure 4 and related text: frequency and percentage of what? the mechanics of how they used STP is not clear

An explanation of the use of STP for the construction of Figure 4 has been added to the text, allowing the choice of targets for in silico studies to be explained.

-l573: Table 4 presents the molecular coupling energies with each of the proteins that had a reported crystal (PDB database) are shown,: too many verbs here!

-l723: however. : however what?

The title of table 4 was rewritten and the text was connected.

However, this is not the main problem with the paper. Outstandingly, in their answer to my objections, the authors seem to understand my point about table 4, however this empathy does not make itself into the paper. Instead they go at length into discussing how table 4 purports a long range of quantitative deductions: what would displace what and so on, with no appreciation or mention of the accuracy of the reported data or comparison with available experiments. May I at least make them notice at least three oddities?

Ki or IC50 values have been added to compare for each of the targets studied in molecular docking to determine what relationship the essential oil components could have.

  1. most of the compounds in their oil bind M1 better than the reference.

Many of the effects that have been reported for this essential oil are related to M1, so there is an interesting relationship for further studies, although they have good energy, functioning as agonists or antagonists depends to a large extent on the interactions and the concentration.

  1. worse, more than half the compounds in their oil bind at least 5 receptors more than the reference ones. is this the receipt for a kind of panacea or a poison?

Indeed, the effect can be from beneficial to poison, it will depend on the dose and therefore its use in a controlled manner, this last part has been included in the conclusions as a warning.

  1. geranial, shall we really discuss it? At any rate I tested it on STP and even other similar software like PLATO and they do not give any credit to it as a plausible M1 target: actually M1 is at the seventh place and no target anyway exhibits a significant probability that geranial is a binder of interest.

            STP and PLATO are platforms that allow an analysis by structural similarity, in the manuscript it was proposed as a platform to determine the set of metabolites identified proteins with which they could interact, however, this is due to similarity, hence the proposal to study now by docking molecular, remembering that they are theoretical studies and therefore are limited, that is why using more than one type of study, and although it is not a priority in STP el Geranial, in molecular docking it presents a good interest opening to future study.

There are interesting data in the paper, and even some potentially interesting compounds like 1-Benzyloxy-9-(phenylthio)-3,7,11,15-tetramethyl-2,6,10,14-hexadecatetraene. The problem is that the conclusions are not warranted by the methods, and this is mostly due to the authors' excessive reliance on their numbers. I am not able to discern the origin of such errors, but it is rather clear that even the relative error in the binding energies intra columns is at least 2 kcal/mol. Certainly it is much more among the different columns, where some data are in line thermodynamically with experimental data while others are off by at least 4 kcal/mol, as I showed in the first reviewing round. It is easy to spot, for example in the large difference of binding energies in the references for M1 and M2.

The use of bioinformatics tools allows the detection of compounds that have not been previously studied to guide new studies, with respect to shell energies, the calculation made results from the sum of interactions presented by each of the molecules, although there is a direct relationship between the Ki and this calculation there are other factors in the calculation from them, it would be necessary to continue with more studies, including carrying out a QSAR to determine the relationship and determine other factors that influence their calculation. The comment is interesting. It is appreciated and it was considered to make that point clear.